# Investigation of Prognostic Value of Claudin-5, PSMA, and Ki67 Expression in Canine Splenic Hemangiosarcoma

**DOI:** 10.3390/ani11082406

**Published:** 2021-08-14

**Authors:** Juliana Moreira Rozolen, Tamires Goneli Wichert Teodoro, Renata Afonso Sobral, Felipe Augusto Ruiz Sueiro, Renee Laufer-Amorim, Fabiana Elias, Carlos Eduardo Fonseca-Alves

**Affiliations:** 1Department of Veterinary Surgery and Animal Reproduction, Sao Paulo State University—UNESP, Botucatu 18618-681, Brazil; juliana.rozolen@unesp.br; 2Department of Veterinary Clinic, Sao Paulo State University—UNESP, Botucatu 18618-681, Brazil; tamires.teodoro@unesp.br (T.G.W.T.); renee.laufer-amorim@unesp.br (R.L.-A.); 3ONCO CANE-Veterinary, São Paulo 04084-002, Brazil; renatasobral@oncocane.com; 4VetPat Veterinary Laboratory, Campinas 13073-022, Brazil; contato@vetpat.com; 5Veterinary School, Federal University of Fronteira Sul—UFFS, Realeza 85770-000, Brazil; elias.fabiana@gmail.com; 6Institute of Health Sciences, Paulista University—UNIP, Bauru 17048-290, Brazil

**Keywords:** endothelial splenic neoplasia, prostate-specific membrane antigen, canine spleen neoplasia, proliferative index, Ki67

## Abstract

**Simple Summary:**

Canine splenic hemangiosarcoma (HSA) is an aggressive cancer that originates from endothelial cells. In clinical practice, it is commonly noted among canine patients with ruptured tumors, inducing internal bleeding. Although it is highly important in veterinary medicine, very limited information regarding HSA prognostic markers is available. Thus, this study aimed to evaluate the prognostic value of Claudin-5, prostate-specific membrane antigen (PSMA), and Ki67 in HSA-affected canine patients. We evaluated Claudin-5 and PSMA gene and protein expression, assessed the Ki67 index, and compared these with patients’ clinical data. We identified an association between Claudin-5 expression and metastatic status. Patients with higher Claudin-5 expression developed metastasis, and there was an association between PSMA expression and overall survival. Our results suggest that these proteins could be useful prognostic markers for patients with HSA.

**Abstract:**

Splenic hemangiosarcoma (HSA) is a malignant tumor of endothelial cells that affects middle-aged and elderly dogs and is characterized by the formation of new blood vessels, commonly associated with necrotic and hemorrhagic areas. Despite its importance in veterinary medicine, few studies have identified markers with prognostic value for canine HSA. Thus, this study aimed to associate the clinicopathological findings (prostate-specific membrane antigen [PSMA], Claudin-5, and Ki67 gene and protein expression) with overall survival in HSA-affected patients. Fifty-three formalin-fixed and paraffin-embedded canine splenic HSA samples, previously diagnosed by histopathological examination, were used in this study. Claudin-5, PSMA, and Ki67 protein expression levels were evaluated by immunohistochemistry, and gene expression was evaluated by quantitative polymerase chain reaction. Claudin-5 protein overexpression was observed in patients with metastasis (*p* = 0.0078) and with stage III tumors compared to those with stage I and II tumors (*p* = 0.0451). In patients treated with surgery alone, low PSMA gene and protein expression (*p* = 0.05 and *p* = 0.0355, respectively) were associated with longer survival time. Longer survival time was observed in patients with a low Ki67 index (*p* = 0.0488). Our results indicate that Claudin-5 protein expression is associated with metastatic status, and PSMA gene and protein expression, and Ki67 index are associated with survival time.

## 1. Introduction

Canine splenic hemangiosarcoma (HSA) is a highly aggressive endothelial neoplasm with a high metastatic rate and poor outcome. It is one of the most aggressive and metastatic cancers in dogs [1,2,3,4,5]. It affects middle-aged to elderly dogs, with a greater predisposition for males [1,2,3,4,5,6,7,8]. The most common metastatic sites are the liver and lungs, and less frequently the omentum, kidneys, retroperitoneum, and skin [5,6,7,8,9,10,11,12,13]. Despite the importance of this tumor in clinical practice, few advances have been achieved recently regarding prognosis and treatment of this tumor subtype [6]. Splenectomy is the gold standard treatment for splenic has. However, due to poor outcomes, adjuvant chemotherapy is associated with controversial results [5,6,7,8,9,10,11,12,13].

The current literature regarding canihasHSA is composed of retrospective studies or studies evaluating different markers in histological/necropsy samples, without association with survival. There are a limited number of studies investigating the prognostic value of different markers in chasne HSA. By reviewing the previous literature, we identified Claudin-5 [14,15,16,17] and prostate-specific membrane antigen (PSMA) [18,19] as potential markers ohasanine HSA. Claudin-5 is a tight junctional membrane protein responsible for the integrity and maintenance of vascular membranes [15,16]. Claudin-5 has been detected in the membranes of endothelial neoplastic cells in all histological typehasf canine HSA, and researchers have proposed Claudin-5 as a diagnostic marker [17]. However, diagnosing hasine splenic HSA is usually challenging.

PSMA is claimed to be a prostatic cancer marker and has been detected in biological fluidshas patients with HSA. Previously, PSMA expression was readily identifiable in hemorrhagic effusions; this has potential value for the early detection through molecular diagnostihasests for presence HSA cells in hemorrhagic effusion [18]. Since PSMA is expressed in canine prostate, it is not exclusive to malignant endothelial cells and not considered a diagnostic marker [18,19,20]. Although PSMA was previously studied in hemorrhagichasfusions of dogs with HSA, no previous studies have evaluated the association between the expression of PSMA in hemangiosarcoma tissues and the survival time of patients.

Among the markers used in veterinary oncology, Ki-67 is a nuclear protein widely used for the evaluation of cell proliferation. This protein is expressed only in proliferating cells and in all active phases of the cell cycle, including early division, and observed mainly in patients at a late stage with poor prognosis [21]. Besides its importance in veterinary oncology, to the best of our knowledge, no previous study has reported the prognostic value of Ki67 in canine HSA. Due to the limited information regarding markers with prognostic value for canine HSA, our research aimed to examine the association of Claudin-5, PSMA, and Ki67 expression with the clinical and pathological findings of splenic HSA-affected dogs.

## 2. Materials and Methods

### 2.1. Study Design

This was a retrospective non-randomized study including 53 canine splenic HSA cases from Sao Paulo State University and the OncoCane Oncology Center. The inclusion criteria for this study were as follows: histopathological analysis confirming splenic HSA diagnosis, presence of complete clinical information, and availability of paraffin blocks from the primary tumor. Only patients with imaging examinations (abdominal ultrasound and three-view thoracic radiography) to perform tumor staging were included in this study. Patients without clinical information or with other primary tumors at diagnosis, and those with tumors without internal Ki67-positive expression were excluded. A total of 53 out of 111 samples met the inclusion criteria. Since castration is not common in Brazil, our study population is based on non-spayed dogs (*n* = 53).

The clinical stage was performed according to the tumor size (T), presence of lymph node involvement (N), and distant metastasis (M) based on a modification of the World Health Organization scheme [9].

### 2.2. Morphological Analysis

A new section was obtained from the paraffin blocks and stained with hematoxylin and eosin. Histological and histological subtype evaluations were performed using an optical microscope by two collaborators (J.M.R and C.E.F-A), according to the methods of Göritz et al. [10]. The morphological analysis was performed in a blinded fashion, and clinical information was omitted during analysis.

Besides that, for statistical proposes we grouped samples with vascular-like histological subtypes (cavernous and capillary) as well-differentiated subtype and the solid pattern as undifferentiated pattern.

### 2.3. Gene Expression

Transcript levels of Claudin-5 and PSMA were evaluated; tissues fixed in formalin and embedded in paraffin were used for mRNA extraction, according to a previous procedure [11]. For mRNA extraction, five 6-µM tissue sections were placed on histological slides and tumor areas were macrodissected using a scalpel. mRNA was extracted using a commercial kit (RecoverAll™ total nucleic acid kit, Ambion Life Technologies, Carlsbad, MA, USA) according to the manufacturer’s instructions. mRNA concentration was determined using a spectrophotometer (Nano Drop™, ND-8000, Thermo Scientific, Carlsbad, MA, USA). cDNA was synthesized at a final volume of 20 µL, and each reaction mixture contained 1 µg total RNA treated with DNAse I (Life Technologies, Rockville, MD, USA), 200 U SuperScript III reverse transcriptase (Life Technologies), 4 µL 5× SuperScript First-Strand, 10 mM dNTPs (Life Technologies), 1 µL Oligo-(dT) 18 (500 ng/µL) (Life Technologies), 1 µL random hexamers (100 ng/µL) (Life Technologies), and 1 µL 0.1 M DTT (Life Technologies). Reverse transcription was performed for 60 min at 50 °C, and the enzyme was inactivated for 15 min at 70 °C.

Six endogenous genes were used (*ACTB*, *GAHPD*, *HPRT*, *RPL8*, *RPS19*, and *PRS5*), and the forward and reverse primers for all genes are shown in Appendix A. The reaction was conducted in a total volume of 10 µL containing Power SYBR Green PCR Master Mix (Applied Biosystems, Foster City, CA, USA), 1 µL cDNA (1:10), and 0.3 µM of each primer pair, in triplicate, using QuantStudio 12 K Flex Thermal Cycler equipment (Applied Biosystems). A dissociation curve was calculated for all experiments to determine the PCR product specificity. All experiments were performed in triplicate. mRNA from five hematomas was extracted and used for normalization.

### 2.4. Antibody Validation Statement

Since most of the antibodies used are produced to react with human tissue, we verified the cross-reactivity of all antibodies with canine tissue. Reactivity metrics of the Ki-67 antibody (Clone MIB-1; Dako Cytomation, Carpinteria, CA, USA) with canine tissue were provided by the manufacturer. Claudin-5 was previously validated by Jakab et al. [17] and the cross-reactivity of PSMA with canine tissue was previously reported by Dowling et al. [18].

### 2.5. Immunohistochemistry

Immunohistochemistry was performed using a polymer system and 3,3′-diaminobenzidine (DAB) as a chromogen. Initially, Ki67 staining was performed in all paraffin blocks (*n* = 53), and samples without Ki67 expression (internal control) were excluded from this study (*n* = 6). Antigen retrieval was performed in a pressure cooker (Pascal; Dako Cytomation) with citrate buffer (pH 6.0), and endogenous peroxidase blockage was performed using 8% hydrogen peroxide diluted in methyl alcohol for 1 h for both Ki-67 and Claudin-5. For the PSMA antibody, peroxidase blockage was performed for 30 min. A commercial protein block reagent (Dako Cytomation) was used to block non-specific proteins in Table 2.

Mouse monoclonal Ki-67 (Clone: MIB1, Dako Cytomation, Carpinteria, CA, USA) was diluted at 1:50. Mouse monoclonal CLAUDIN-5 antibody (clone: clone 4C3C2, Zymed Inc., San Francisco, CA, USA) and mouse monoclonal PSMA (clone: OTI3H5, Novus Biologicals, Littleton, CO, USA) were diluted at 1:200 and 1:2000, respectively. Slides were then incubated with Dako EnVision™ + Dual Link System-HRP secondary antibody and the chromogen IP FLX DAB (Dako Cytomation) for 1 h. Slides were counterstained with hematoxylin. Internal blood vessels of normal prostatic tissue were used as positive controls for Claudin-5 and the luminal cells used as control for PSMA. For negative controls, primary antibodies were replaced with isotype immunoglobulins at the same concentration as the respective primary antibodies. The protocol information is shown in Appendix A.

For immunohistochemistry analysis, a qualitative evaluation describing each marker immunolocalization was performed. Immunohistochemical expression was evaluated using ImageJ software (ImageJ v1.47; NIH, Bethesda, MD, USA) according to the methods of Da Silva et al. [21]. Briefly, a threshold tool was determined for Claudin-5 and PSMA expression based on positively stained cells. Analysis was then performed in five fields (40×) for each sample, and mean expression was used as the final value. The threshold values for Claudin-5 were 1–255 (hue), 15–255 (saturation), and 100–175 (brightness). For PSMA, the threshold values were 1–255 (hue), 30–255 (saturation), and 100–180 (brightness). The evaluation of Ki67 was performed by manually counting at least 1000 cells (positive and negative) in five high-power fields (400×). The number of positive cells divided by the total number of cells was determined as the percentage of positive cells. The final Ki67 index was determined as the mean of five fields.

### 2.6. Statistical Analysis

For statistical purposes, we grouped samples according to clinicopathological criteria and compared the different categorical variables with overall survival (OS). The categorical variables included histological subtypes, metastatic status, clinical stage, and treatment. The immunohistochemical and gene expression results were evaluated by their mean values and then divided into two groups (low or high expression). The segregation of low versus high expression for gene expression data was performed after normalizing the data using mRNA from hematomas. Samples with expression equal to or lower than the mean value were classified as low expression, and samples with expression higher than the mean value were classified as high. Since adjuvant treatment can affect the OS of patients [2,6,9], we evaluated samples from canine patients that underwent only surgery and surgery associated with chemotherapy separately. Moreover, among the patients that underwent chemotherapy, we grouped patients treated with anthracycline-based chemotherapy and metronomic cyclophosphamide. Overall, adjuvant treatment can affect the OS of patients. To evaluate the correlation among all variables, a matrix of multiple correlations was performed using Spearman’s test. Survival analysis was performed using the log-rank test and Kaplan–Meier curves. Survival data were censored when dogs were still alive or had died from other causes. Statistical analysis was performed using GraphPad Prism v8.1.0 (GraphPad Software Inc., La Jolla, CA, USA). The survival information was censored when patients were still alive or died from other causes. Statistical significance was considered when *p* values were <0.05.

## 3. Results

### 3.1. Clinical Data

The mean age of these patients was 10.7 years (±2.0 years); the sample included 79.24% (*n* = 42) females and 20.75% (*n* = 11) males. Mixed-breed dogs were the most frequent (22/53), followed by Pit Bull (7/53), Cocker Spaniel (3/53), Dachshund (3/53), Golden Retriever (3/53), Border Collie (2/53), Boxer (2/53), German Shepherd (2/53), Poodle (2/53), and Belgian Shepherd (2/53). There was one of each of Brazilian Mastiff, Labrador, Lhasa Apso, Schnauzer, and Yorkshire breeds. The presence of metastasis at diagnosis was seen in 60.38% (32/53) of cases, and the most prevalent sites were the liver (18/32), lungs (6/32), skin (6/32), and heart myocardium (2/32).

The mean OS was 155.94 days (±148.37 days), independent of patient treatment (Figure 1A). Patients with metastasis at the time of diagnosis had a poor prognosis (*p* < 0.0001) (Figure 1B). Regarding the clinical stage, 60.3% (33/53) were at stage III, 26.4% (14/53) were at stage II, and 13.3% were at stage I (7/53). There was an inverse relationship between OS and tumor stage. Thus, patients at later stages had shorter survival times (*p* = 0.0444) (Figure 1C). Regarding treatment, 66% (35/53) of patients underwent surgery as the only treatment option (*n* = 35), and 27% (14/53) were treated with surgery associated with adjuvant chemotherapy (*n* = 14). In addition, in 7% (4/53) of cases, the owner declined any treatment. Patients who only underwent surgery experienced an average survival time of 128.83 (±118.84) days. In contrast, those who underwent surgery associated with chemotherapy experienced a longer survival time (262.43 ± 243.53 days). The patients without treatment experienced the shortest survival time (20.50 ± 19.5 days) (Figure 1D).

### 3.2. Histopathological Analysis

Tissue samples were classified as cavernous (23/53), capillary (16/53), and solid (14/53), and there was no association between survival and different histological subtypes (*p* = 0.1286). Based on the degree of differentiation, samples were also classified as differentiated (39/53) and undifferentiated (14/53) for sample groupings and to examine the association with survival time. However, there was no association between survival time and the degree of tumor differentiation (*p* = 0.3837).

### 3.3. Gene Expression

The mean relative quantification (RQ) of Claudin-5 was 3.36 (±2.9) and there was no statistical difference in Claudin-5 gene expression between the metastatic and non-metastatic samples (*p* = 0.8266). Regarding the tumor stage, there was no statistical difference in OS among the three stages (*p* = 0.4804) or between stages I and II (*p* = 0.3011), stages I and III (p = 0.5142), or stages II and III (*p* = 0.4448). Claudin-5 expression was also not associated with different histological subtypes. When comparing Claudin-5 expression among capillary, cavernous, and solid subtypes, there was no statistical difference (*p* = 0.2959). Claudin-5 expression did not differ between capillary and cavernous (*p* = 0.6573), capillary and solid (*p* = 0.4252), or cavernous and solid tissues (*p* = 0.1338). The other comparisons did not show any association between Claudin-5 expression and other clinicopathological parameters. The results of all comparisons are shown in Table 1.

The mean RQ of PSMA was 5.26 (±5.19). There was no statistical difference in PSMA gene expression between metastatic and non-metastatic samples (*p* = 0.1555). Regarding tumor stage, there was a statistical difference between the three stages (*p* = 0.041), with patients with stage III tumors having higher PSMA levels. There was no statistical difference in PSMA expression between tumors at stages I and II (*p* = 0.1807) or stages I and III (*p* = 0.9699). In contrast, samples from patients with stage III tumors had higher PSMA expression than samples from patients with stage II tumors (*p* = 0.0319). When the association between histological subtypes and PSMA gene expression was evaluated, a higher PSMA expression was observed in the cavernous subtype compared with the capillary subtype (*p* = 0.046) and in the cavernous subtype compared with the solid subtype (*p* = 0.05). There was no statistical difference between the capillary and solid subtypes (*p* = 0.9497). Although PSMA was associated with tumor stage and histological subtype, no association was found between PSMA expression and OS (*p* = 0.4080). These data are shown in Table 1.

### 3.4. Claudin-5, PSMA, and Ki67 Immunoexpression

Claudin-5 showed membranous and cytoplasmic staining (Figure 2A–C) in endothelial neoplastic cells and pre-existing blood vessels. PSMA showed cytoplasmic staining (Figure 2D–F) only in neoplastic cells, and Ki-67 was positive in the nucleus of endothelial neoplastic cells. Claudin-5 protein showed a mean expression of 12% (±5.73%) among HSA samples and Claudin-5 protein overexpression was found in samples from patients with metastasis at the time of diagnosis (*p* = 0.0078). Thus, higher Claudin-5 expression was observed in metastatic samples than in non-metastatic samples. We also identified an association between Claudin-5 expression and tumor stage. Claudin-5 expression was higher in samples from patients at stage III than in those at stages I and II (*p* = 0.0451) (Figure 3). A representative image from all antibodies in each histological subtype (capillary, cavernous and solid) was provided in Figure 4.

The association of Claudin-5 expression with different histological subtypes was investigated, and no significant statistical difference was observed among capillary, cavernous, and solid subtypes (*p* = 0.2460). The other comparisons did not show significant statistical differences (Table 2).

The mean expression of PSMA was 5% (±3.89%) in HSA samples, and there was no association between PSMA protein expression and patients’ metastatic status (*p* = 0.5650). Regarding tumor stage, there was no statistical difference among the three stages and PSMA expression (*p* = 0.2804). Comparison between two different stages was made, and there was no statistical difference when comparing tumor stages I and II (*p* = 0.1455), I and III (*p* = 0.1931), or II and III (*p* = 0.5401). PSMA protein expression did not differ among the three histological subtypes (*p* = 0.2653) or when comparing capillary and cavernous (*p* = 0.1685), capillary and solid (*p* = 0.7336), or cavernous and solid tissues (*p* = 0.2022). When differentiated and undifferentiated patterns were compared, no statistically significant difference was observed (*p* = 0.3372). The results of all comparisons between PSMA protein expression and clinicopathological findings are shown in Table 2.

Ki-67 expression was evaluated in 47 out of 53 samples because in six samples it was not possible to evaluate 1000 cells. The mean Ki67 expression was 16% (±14.06%), and there was no association between Ki67 expression and metastatic status (*p* = 0.8323). In contrast, an association was identified between Ki67 index and tumor stage, with tumor stages II and III showing a higher Ki67 index when compared to stage I (*p* = 0.0144). A higher Ki67 index was observed in tumor stages II (*p* = 0.0033) and III (*p* = 0.0355) when they were individually compared with stage I. In contrast, there was no statistical difference between tumor stages II and III (*p* = 0.1222). The solid subtype showed a higher proliferative index than the capillary subtype (*p* = 0.0433). A higher proliferative index was also observed in undifferentiated tumors than in differentiated HSA (*p* = 0.05) (Table 2).

### 3.5. Prognostic Associations

We included all survival time associations (prognostic value) in our analyses, due to the separation of patients treated only with surgery or treated with surgery and chemotherapy. The mean gene expressions of Claudin-5 in samples from patients treated with surgery alone and patients treated with surgery and chemotherapy were 2.478 ± 2.4 (*n* = 35) and 5.3 ± 5.6 (*n* = 14), respectively. For Claudin-5 protein expression, the mean expressions for patients that underwent only surgery and patients treated with surgery and chemotherapy were 10.6 ± 6.9 and 14.1 ± 6.1, respectively. There was no significant statistical difference between Claudin-5 low or high gene expression in patients treated with surgery (*p* = 0.424) or surgery associated with chemotherapy (*p* = 0.8454). There was also no statistical difference in Claudin-5 expression in samples from patients who underwent surgery (*p* = 0.694) or surgery associated with chemotherapy (*p* = 0.4001).

Samples from patients who underwent surgery showed a mean PSMA gene expression of 10.6 ± 6.9, and samples from patients treated with surgery and chemotherapy showed a mean gene expression of 14.1 ± 7.4. Mean *PSMA* gene expression was 4.8 ± 3.9 in samples from patients treated with surgery and 4.8 ± 4.3 in samples of patients treated with surgery and chemotherapy. There was no significant statistical difference between PSMA gene expression and OS in patients who underwent surgery. In contrast, among those who underwent chemotherapy, patients with low PSMA expression had a longer survival time (*p* = 0.05). For patients treated with surgery, low PSMA protein expression was associated with increased survival time (*p* = 0.0355) (Figure 5A). However, when the patients were treated with chemotherapy, the opposite relationship was observed. Patients with lower PSMA protein expression had shorter survival times (*p* = 0.0113) (Figure 5B).

The mean Ki67 proliferative index in patients treated with surgery was 20.8 ± 19.5 and in those submitted to surgery and chemotherapy was 3.4 ± 4.7. A longer survival time was observed in patients who underwent surgery with a low Ki67 index (*p* = 0.0488) (Figure 5C). A longer survival time was also observed in patients with a low Ki67 index in the group of patients treated with chemotherapy (*p* = 0.0143) (Figure 5D).

### 3.6. Matrix of Multiple Correlations

Multiple correlation analysis revealed a strong positive correlation between tumor stage and development of metastasis (r = 0.9597). Thus, patients at higher stages were correlated with metastasis at different sites at the time of diagnosis. A positive correlation was also observed between Claudin-5 protein expression and OS. In contrast, patients with higher Claudin-5 protein expression were also correlated with longer survival time (r = 0.4382). The Ki67 index was negatively correlated with Claudin-5 protein expression (r = 0.4264). The matrix of multiple correlations for all markers is shown in Figure 6.

## 4. Discussion

Canine HSA is one of the most common and metastatic cancers in dogs [1,2,3,4,5]. In our study, the average age of the patients was 10.7 ± 2.0 years and females were more affected than males, representing 79.24% of our samples. Metastasis occurred in 60.4% of cases, with the liver, lungs, and skin being the most affected organs, in agreement with previous studies [5,7,10,11,12,13,22,23]. The mean OS was 155.94 days (±148.37 days), regardless of treatment. In total, 66% of patients underwent surgery and had an average survival time of 128.83 days (±118.84 days), and 27% of patients underwent surgery and chemotherapy and had an average survival time 262.43 days (±243.53). Seven percent of patients did not receive any treatment and had an average OS of 20.50 days (±19.5), which is also consistent with previous studies [2,5,24,25,26,27,28]. However, this data should be evaluated carefully since the previous studies are retrospective and include patients from different stages and other inclusion criteria.

In our study, stage III tumors were the most frequent, confirming advanced disease at diagnosis. Since the spleen is an internal organ, it is commonly diagnosed in the late stages. Our survival data also demonstrated a shorter survival time for patients with stage III tumors, in agreement with previous studies showing that the majority of splenic HSA present aggressively at stage III with decreased survival time due to the presence of metastasis. Tumor staging is one of the most important prognostic factors for all subtypes of tumors, and we showed a shorter survival time in patients with stage III tumors than in patients with stage I and II tumors. Through our matrix of multiple correlations, we confirmed the importance of tumor stage in predicting metastatic disease. This finding has been widely reported in the literature [1,2,5,23,29].

Previous studies have shown that stage III tumors have the worst prognosis, and patients at stages I and II present a challenge [23,30]. In addition, patients who underwent chemotherapy associated with surgery had a longer survival time than those who underwent surgery alone. Since chemotherapy could have a direct impact on overall survival, we opted to analyze our overall survival data, grouping separate patients who underwent surgery only from patients who underwent surgery associated with chemotherapy [27,28,31]. Due to the retrospective nature of our research, for those who underwent surgery and chemotherapy, we used different protocols. Patients treated with anthracycline-based chemotherapy were grouped with patients treated with metronomic cyclophosphamide. We used this grouping because we had a low number of patients treated with chemotherapy, and previous literature has provided sufficient evidence to support this approach. Overall, there was no difference in OS in patients treated with anthracycline-based chemotherapy or metronomic cyclophosphamide [27,28,31]. Although chemotherapy seems to improve patient survival [5,27,28,31], this information is based on retrospective studies with different canine populations and non-randomized enrollment. In our study, patients that underwent only surgery had a higher survival mean compared with patients submitted to surgery and chemotherapy. However, we have a high number of censored patients in the group of patients treated only with surgery. Moreover, reviewing the previous studies in the literature, different protocols were applied, and any conclusion should be drawn carefully [5,27,28,31].

Regarding the markers used in our study, Claudin-5 is a member of the Claudin protein family and may be directly or indirectly related to tumor motility and neovascularization, facilitating metastasis and tumor progression because it is closely linked to epithelial and endothelial cells [17,32]. We did not find an association between Claudin-5 gene expression and the evaluated parameters. In contrast, Claudin-5 protein was overexpressed in samples from patients with metastasis. Claudin-5 expression was also associated with stage III tumors. Thus, in more advanced stages of the disease, the expression of Claudin-5 was more evident. A previous study [17] reported that the high expression of Claudin-5 occurs mainly in solid and undifferentiated histological types. In our study, we found high expression in the cavernous and differentiated histological subtypes. Claudin-5 could be used as a marker with greater sensitivity and could also be of diagnostic value in the differential diagnosis of canine HSAs from other sarcomas with hemorrhage or increased vascularization [16,17,32,33,34]. Claudin-5 might also represent an interesting prognostic marker for canine HSA metastasis. For this purpose, further studies are needed to establish its potential and determine expression parameters.

According to previous literature [18,19], PSMA gene expression is predictive of HSA occurrence and metastasis development. In our study, we observed high PSMA gene expression in patients with metastasis (stage III) and cavernous histological type. Interestingly, in samples from patients that underwent surgery and chemotherapy treatment, a longer survival time was observed in patients with low PSMA gene expression. There are few studies that have reported a correlation between PSMA and HSA survival. Relative expression of PSMA could be used as a prognostic marker. Although PSMA has been previously associated with canine HSA abdominal effusion, we demonstrated a direct relationship between PSMA gene expression and patient prognosis. One of the most challenging tasks in using gene expression for this analysis is using a normalization control for qPCR experiments. Although canine HAS originates from endothelial cells, there are no established normal baselines for gene expression experiments in HSA. Dowling et al. [18] investigated hemorrhagic body cavity effusions in HSA, extracting endothelial neoplastic cells and carrying out gene expression experiments using normal endothelial cells as controls. However, normal endothelial cells from HSA tissue samples from patients could show alterations in their expression patterns under culture conditions and do not represent reliable controls.

Previous experiments using gene expression have compared tumors with different histological or molecular patterns. Tamburini et al. [35] assessed HSA gene expression by microarray, and RT-PCR analysis as a comparison, using mRNA from hematomas. Thus, our choice to use mRNA from splenic hematomas was based on these previous results.

Although Ki67 expression is widely seen in canine tumors, we did not find any previous studies evaluating the Ki67 proliferative index in canine splenic HSA. In our study, long survival time was correlated with undergoing at least one of the proposed treatments; thus, long-term survival was characterized by a low Ki67 index. Previously, Moore et al. [29] studied 30 cases of canine HSA and identified a correlation between mitotic index and OS. Our immunohistochemical results demonstrated that patients that underwent one of the proposed treatments had a lower rate of proliferation by Ki67 analysis and had longer survival; a high Ki67 index was found in a patient with a stage II tumor (solid and undifferentiated histological type), again corroborating the results of previous studies.

All markers studied demonstrated strong and efficient staining in patients that had metastasis and more advanced stages, with the cavernous type being the most affected due to the arrangement of vascular channels, and the proliferative index accompanied tumor progression. Thus, we propose that Claudin-5, PSMA, and Ki67 could be used as a prognostic panel for patients with HSA.

## 5. Conclusions

Overall, our study confirmed that patients with stage III and metastatic tumors have poor prognosis. Adjuvant chemotherapy provides a higher OS for patients and should be indicated for all patients with splenic HSA. Moreover, our results indicate that Claudin-5 protein expression is associated with metastatic status, and PSMA gene and protein expression and Ki67 index are associated with survival time, making them potential prognostic markers for canine HSA.

## Figures and Tables

**Figure 1 animals-11-02406-f001:**
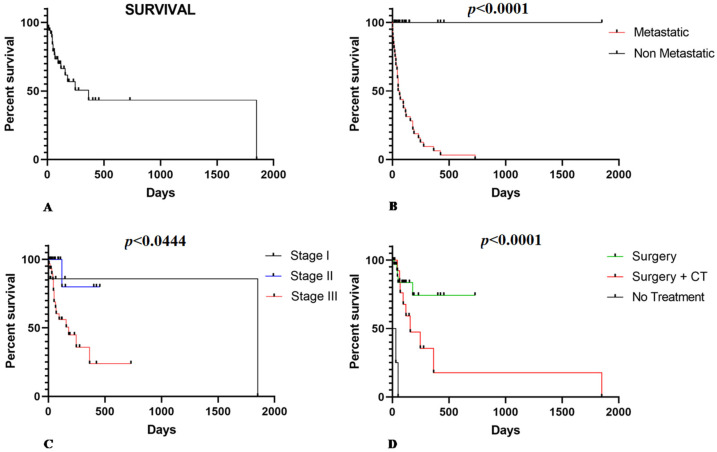
(**A**) Overall survival in HSA-affected dogs according to clinical pathological criteria. (**B**) Patients with metastatic splenic HSA experienced a shorter survival time (*p* < 0.0001). (**C**) Patients at stage III show shorter survival time than stage I and II patients (*p* = 0.0444). (**D**) Patients that received surgical or surgery associated with adjuvant chemotherapy (CT) as treatment show a longer survival time than those that did not receive treatment or only surgery treatment (*p* < 0.0001).

**Figure 2 animals-11-02406-f002:**
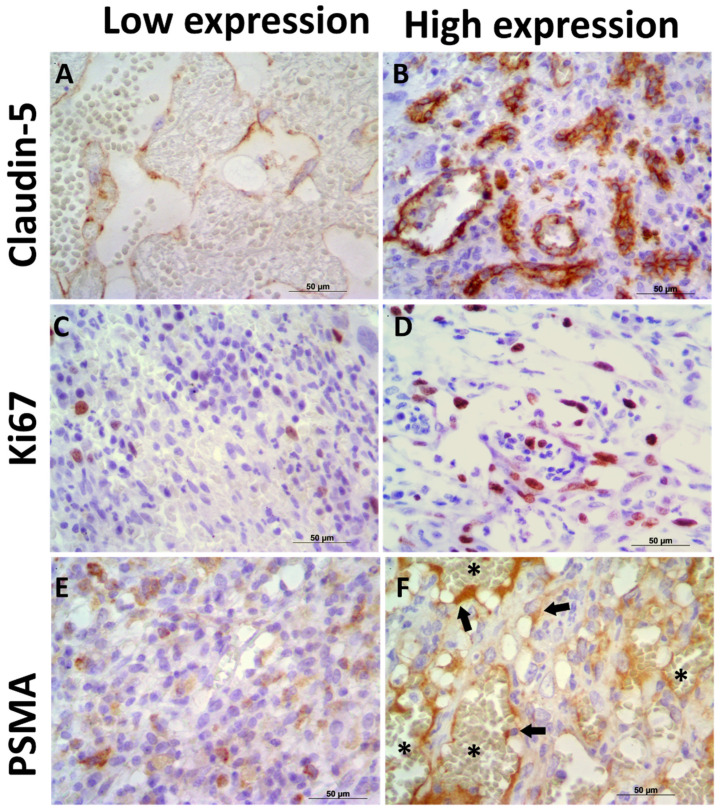
Immunoexpression of Claudin-5, PSMA, and Ki67 in canine hemangiosarcoma (HSA) samples. (**A**) Low Claudin-5 immunoexpression in a capillary HSA sample. (**B**) Canine HSA showing high Claudin-5 expression in neoplastic endothelial cells. (**C**) Low Ki67 expression in a solid HSA sample. (**D**) High Ki67 expression in a canine HSA sample. (**E**) Low PSMA expression in a solid HSA sample. (**F**) High PSMA expression in neoplastic endothelial cells (arrows) in a capillary HSA sample. Note the negative expression in red cells (asterisk), confirming satisfactory endogenous peroxidase blocking. Harris hematoxylin counterstain, 40×.

**Figure 3 animals-11-02406-f003:**
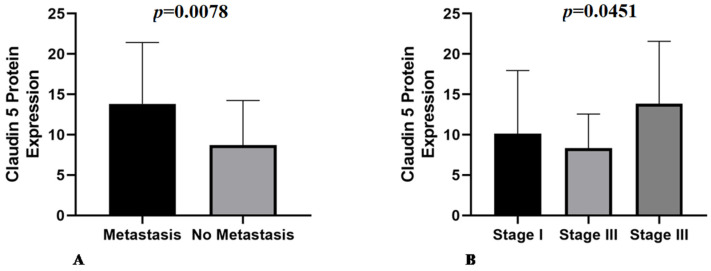
(**A**) Metastatic samples have higher Claudin-5 expression compared with non-metastatic samples (*p* = 0.0078). (**B**) Association of Claudin-5 expression with tumor stage, with stage III tumors showing higher expression than stage I and II tumors (*p* = 0.0451).

**Figure 4 animals-11-02406-f004:**
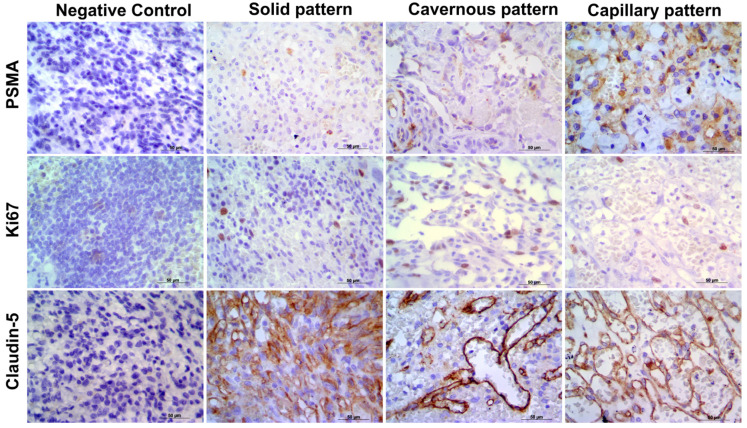
Representative image of each antibody in capillary, cavernous and solid patterns. It is possible to observe the negative control for each antibody and PSMA, Ki67, and Clauding-5 expression in the different histological subtypes. Harris hematoxylin counterstaining, DAB, 40.

**Figure 5 animals-11-02406-f005:**
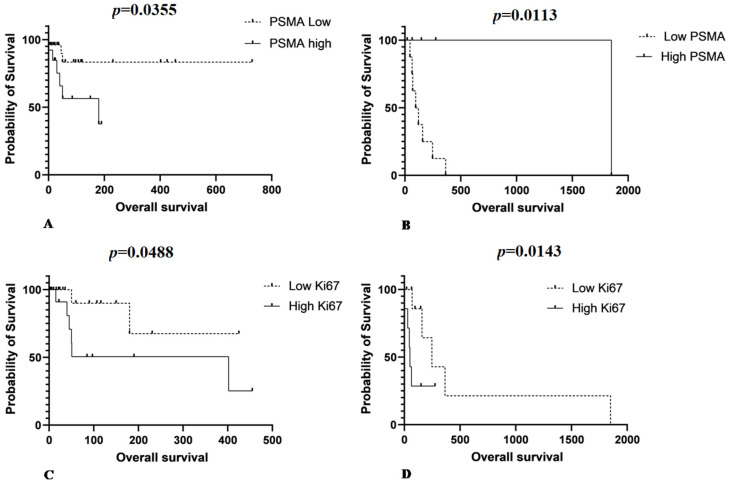
(**A**) Patients that underwent surgery as single therapy show low PSMA protein expression associated with increased survive time (*p* = 0.0355). (**B**) Patients treated with surgery associated with chemotherapy show the opposite relationship between PSMA and overall survival. Patients with low PSMA protein expression experience a shorter survival time (*p* = 0.0113). (**C**) A longer survival time is observed in patients treated with surgery with a lower Ki67 index (*p* = 0.0488) and (**D**) a longer survival time is observed for patients with low Ki67 index in those treated with chemotherapy (*p* = 0.0143).

**Figure 6 animals-11-02406-f006:**
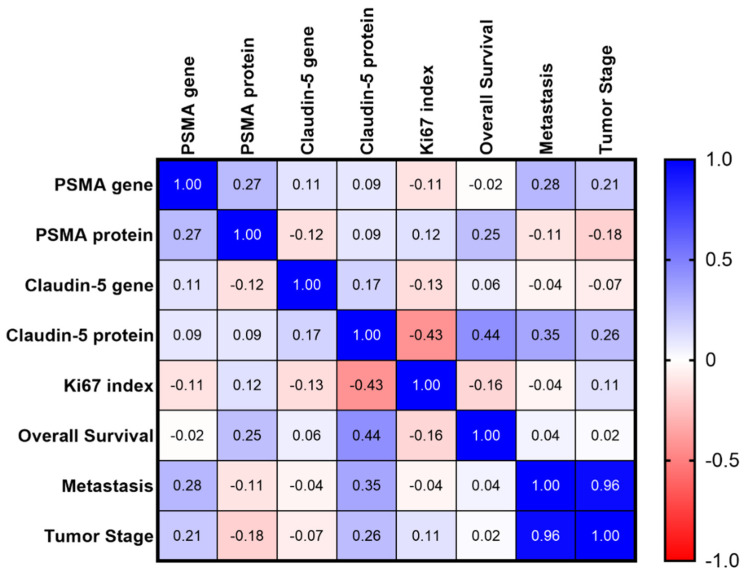
Matrix of multiple correlations for all studied markers. Positive correlations are shown in blue and negative correlations are shown in red. The strength of correlations is depicted by color intensity.

**Table 1 animals-11-02406-t001:** Comparison between gene expression of Claudin-5 and PSMA and clinicopathological parameters.

Clinicopathological Parameter	Claudin-5	PSMA
Relative Quantification	3.36 (±2.9)	5.26 (±5.19)
Overall Survival	*p* = 0.4244	*p* = 0.4080
Metastatic Status	*p* = 0.8266	*p* = 0.1555
Stage	*p* = 0.4804	*p* = 0.041 *
Stage I × II	*p* = 0.3011	*p* = 0.1807
Stage I × III	*p* = 0.5142	*p* = 0.9699
Stage II × III	*p* = 0.4448	*p* = 0.0319 *
Histological Type	*p* = 0.2959	*p* = 0.1815
Capillary × Cavernous	*p* = 0.6573	*p* = 0.046 *
Capillary × Solid	*p* = 0.4252	*p* = 0.9497
Cavernous × Solid	*p* = 0.1338	*p* = 0.05 *
Differentiated × Undifferentiated	*p* = 0.1415	*p* = 0.3285
Treatment	*p* = 0.1684	*p* = 0.3580

* Statistically significant difference.

**Table 2 animals-11-02406-t002:** Association of the different proteins with clinicopathological parameters.

Parameter	Claudin-5	PSMA	Ki-67
Mean Expression	12% (±5.73%)	5% (±3.89%)	16% (±14.06%)
Overall Survival	*p* = 0.9390	*p* = 0.8979	*p* = 0.4542
Metastatic Status	*p* = 0.0078 *	*p* = 0.5650	*p* = 0.8323
Stage	*p* = 0.0451 *	*p* = 0.2804	*p* = 0.0144 *
Stage I × II	*p* = 0.9589	*p* = 0.1455	*p* = 0.0033 *
Stage I × III	*p* = 0.0377	*p* = 0.1931	*p* = 0.0355 *
Stage II × III	*p* = 0.0152 *	*p* = 0.5401	*p* = 0.1222
Histological Types	*p* = 0.2460	*p* = 0.2653	*p* = 0.0942
Capillary × Cavernous	*p* = 0.2897	*p* = 0.1685	*p* = 0.1685
Capillary × Solid	*p* = 0.5439	*p* = 0.7336	*p* = 0.0433 *
Cavernous × Solid	*p* = 0.1196	*p* = 0.2022	*p* = 0.1284
Differentiated × Undifferentiated	*p* = 0.1663	*p* = 0.3372	*p* = 0.05 *
Treatment	*p* = 0.1294	*p* = 0.0805	*p* = 0.0043 *

* Significant statistical difference.

## Data Availability

Data is contained within the article or Appendix A.

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
