# Peer review of "Investigation of Prognostic Value of Claudin-5, PSMA, and Ki67 Expression in Canine Splenic Hemangiosarcoma"

_animals, 2021, doi:10.3390/ani11082406_

Round 1

Reviewer 1 Report

The aim of this study was to examine the association of Claudin-5, PSMA, and Ki67 expression with the clinical and pathological findings in canine splenic hemangiosarcoma (HSA). The authors proposed that Claudin-5, PSMA, and Ki67 could be used as a prognostic panel for patients with HSA and the study “confirmed that patients with stage III and metastatic tumors have a poor prognosis. Adjuvant chemotherapy provides a higher OS for patients and should be indicated for all patients with splenic HSA”.

As strengths I point out the manuscript as a whole. In fact, is very well written, its sections are presented in a balanced, coherent and in an assertive way. The information about the Materials and Methods allows their reproduction by other researchers, and the results are very well presented and supported in figures and tables, including photomicrographs, informative enough. The data are properly discussed and compared with the results of other authors.

Note: I did not have access to supplementary material

Comment 1 (Key words): These keywords do not favor the further query in bibliographic databases of the work when published. I suggest that the authors consider, at least, canine splenic hemangiosarcoma and, if the journal's guidelines allow, Claudin-5, prostate-specific membrane antigen, Ki67.

INTRODUCTION

Assertive and very well written properly framing the problem / study

Comment 2 (line 65): Probably is “PSMA is claimed to be a prostatic cancer marker…” or is “PSMA is claimed to be a prostatic marker…”

MATERIALS AND METHODS

Comment 3 (line 97-98): When the authors say that the observations/evaluations were made by two of the investigators, do the results reflect the average assessment of these two investigators for all parameters? And regarding the correlations with the different tumour stages, was it done blindly?

RESULTS

Comment 4 (Fig 2): As in line 370, the authors report that the expression of Ki-67 was also observed in solid tissues, and as in the figure 2 legend, for the expression of Claudin-5 and PSMA, the authors refer that it is in endothelial cells, in images C and D are also this cell type? At least in C it seems to be in other cell types.

DISCUSSION

Comment 5 (lines 500-505): It is strange to only see the reference to chemotherapy protocols in the Discussion, I think this information should be included in the MM, since it was a grouping decision based on previous literature and is important for the analysis of the Results.

“Patients treated with anthracycline-based chemotherapy were separated from patients treated with metronomic cyclophosphamide. We used this grouping because we had a low number of patients treated with chemotherapy, and previous literature has provided sufficient evidence to support this approach. Overall, there was no difference in OS in patients treated with anthracycline-based chemotherapy or metronomic cyclophosphamide [27,28,31]”

Author Response

Reviewer 1

The aim of this study was to examine the association of Claudin-5, PSMA, and Ki67 expression with the clinical and pathological findings in canine splenic hemangiosarcoma (HSA). The authors proposed that Claudin-5, PSMA, and Ki67 could be used as a prognostic panel for patients with HSA and the study “confirmed that patients with stage III and metastatic tumors have a poor prognosis. Adjuvant chemotherapy provides a higher OS for patients and should be indicated for all patients with splenic HSA”.

As strengths I point out the manuscript as a whole. In fact, is very well written, its sections are presented in a balanced, coherent and in an assertive way. The information about the Materials and Methods allows their reproduction by other researchers, and the results are very well presented and supported in figures and tables, including photomicrographs, informative enough. The data are properly discussed and compared with the results of other authors.

Answer: Dear reviewer, thank you so much for this nice overview of our manuscript and for the positive criticisms and comments for the manuscript improvement. We have followed all suggestion by all reviewers and editors, and we hope this new version it will be more suitable for publication.

Note: I did not have access to supplementary material

Answer: We are sorry that the supplementary material was not uploaded. We have included in this new version.

Comment 1 (Key words): These keywords do not favor the further query in bibliographic databases of the work when published. I suggest that the authors consider, at least, canine splenic hemangiosarcoma and, if the journal's guidelines allow, Claudin-5, prostate-specific membrane antigen, Ki67.

Answer: Dear reviewer, since we can not use the same words in the title and keywords section, we have included canine spleen neoplasia instead canine splenic hemangiosarcoma (that appears in title) and included the other required keywords. Please, found this information in lines 42-43.

INTRODUCTION

Assertive and very well written properly framing the problem / study

Answer: Thank you so much for this kind and positive comment.

Comment 2 (line 65): Probably is “PSMA is claimed to be a prostatic cancer marker…” or is “PSMA is claimed to be a prostatic marker…”

Answer: Thank you for the comment, we modified for “PSMA is claimed to be a prostatic cancer marker…”, as suggested.

MATERIALS AND METHODS

Comment 3 (line 97-98): When the authors say that the observations/evaluations were made by two of the investigators, do the results reflect the average assessment of these two investigators for all parameters? And regarding the correlations with the different tumour stages, was it done blindly?

Answer: Dear reviewer, we highlighted the two investigators for reviewers checking its credentials to evaluate histological samples. We also performed a blinded analysis; a phrase was added to reinforce this point. This information can be found in lines 100-101.

RESULTS

Comment 4 (Fig 2): As in line 370, the authors report that the expression of Ki-67 was also observed in solid tissues, and as in the figure 2 legend, for the expression of Claudin-5 and PSMA, the authors refer that it is in endothelial cells, in images C and D are also this cell type? At least in C it seems to be in other cell types.

Answer: Dear reviewer, this denomination was suggested by the English review editing service. However, doubling checking the information, it was confused. We modified “solid tissues” for “…cavernous and solid histological subtypes”.  The image C refers to endothelial malignant cells in a solid pattern (histological subtype). For this reason, seems different from others (that show capillary or cavernous subtypes).

DISCUSSION

Comment 5 (lines 500-505): It is strange to only see the reference to chemotherapy protocols in the Discussion, I think this information should be included in the MM, since it was a grouping decision based on previous literature and is important for the analysis of the Results.

“Patients treated with anthracycline-based chemotherapy were separated from patients treated with metronomic cyclophosphamide. We used this grouping because we had a low number of patients treated with chemotherapy, and previous literature has provided sufficient evidence to support this approach. Overall, there was no difference in OS in patients treated with anthracycline-based chemotherapy or metronomic cyclophosphamide [27,28,31]”.

Answer: Dear reviewer, we agree with this point, and we included references regarding treatment in methods section. Since this information was used in statistical analysis, we have included a phrase in lines 175-176, with references, reinforcing this sample segregation.

Reviewer 2 Report

Dear authors,

I enjoyed reading your work. I leave my contribution in the sense that the new wording facilitates the reading of the work.

Title

Evaluation of  the prognostic value of Claudin-5, PSMA, Ki67 expression in canine splenic hemangiosarcoma.

Abstract

Page 1, Lines 38, 40 - Our results indicate that Claudin-5 protein expression is associated with metastatic status, PSMA gene and protein expression, and Ki67 index  are associated with survival time.

Materials and Methods

Page 3, Line 88 - what do you mean with "complete clinical information"?

Page 3, Line 91 - Patients without clinical information or with other primary tumors at diagnosis

Page 4, Line 163 - Statistical analysis (please define p value for statistical significance) all p in italic

Results

Page 5, Lines 191,192 were the liver (18/32), lungs (6/32),  skin (6/32), and heart (2/32).

Page 5, Line 230 (Figure 1D) should be previously announced in the text before and not after figure presentation.

Page 10, Line 393 - PSMA gene expression

Discussion

Page 11, Line  476 - with the liver, lungs, and skin (6/32 for both !!!!! )

References

Are references 19 and 20 the same?

19. Chang SS, Reuter VE, Heston WDW, Bander NH, Grauer LS, Gaudin PB. Five different anti-prostate-specific membrane antigen (PSMA) antibodies confirm PSMA expression in tumor-associated neovasculature. Cancer Res [Internet]. 1999 Jul 1;59(13):3192– 617.
8. Available from: http://www.ncbi.nlm.nih.gov/pubmed/10397265 618
20. S.S. C, V.E. R, W.D.W. H, N.H. B, L.S. G, P.B. G. Five different anti-prostate-specific membrane antigen (PSMA) antibodies confirm PSMA expression in tumor-associated neovasculature. Cancer Res. 1999.

Conclusions

The conclusions presented are important, but they are not in agreement with the objective of the work, which is to evaluate the prognostic value of certain markers.

Note: Supplementary files were not made available to the reviewer.

Author Response

Reviewer 2

Dear authors,

I enjoyed reading your work. I leave my contribution in the sense that the new wording facilitates the reading of the work.

Answer: Dear reviewer, thank you so much for such a nice overview of our manuscript and for providing such a positive criticism for our manuscript improvement. We have added all suggestions in the revised version, and hope this new version is more suitable for publication.

Title

Evaluation of the prognostic value of Claudin-5, PSMA, Ki67 expression in canine splenic hemangiosarcoma.

Answer: thank you so much for this suggestion. We have adjusted following this recommendation.

Abstract

Page 1, Lines 38, 40 - Our results indicate that Claudin-5 protein expression is associated with metastatic status, PSMA gene and protein expression, and Ki67 index  are associated with survival time.

Answer: thank you so much for this suggestion. We have adjusted following this recommendation.

Materials and Methods

Page 3, Line 88 - what do you mean with "complete clinical information"?

Answer: Dear reviewer, we meant the availability of all clinical data, including age, breed, gender, clinical signs, response to therapy and any other clinical information.

Page 3, Line 91 - Patients without clinical information or with other primary tumors at diagnosis

Answer: thank you so much for this suggestion. We have adjusted following this recommendation.

Page 4, Line 163 - Statistical analysis (please define p value for statistical significance) all in italic

Answer: thank you so much for this suggestion. We have adjusted following this recommendation.

Results

Page 5, Lines 191,192 were the liver (18/32), lungs (6/32), skin (6/32), and heart (2/32).

Answer: thank you so much for this suggestion. We have adjusted following this recommendation.

Page 5, Line 230 (Figure 1D) should be previously announced in the text before and not after figure presentation.

Answer: The reviewer is completely right. We are sorry for this mistake. We announced the Figure 1D prior its presentation, as suggested.

Page 10, Line 393 - PSMA gene expression

Answer: thank you so much for this suggestion. We have adjusted following this recommendation.

Discussion

Page 11, Line  476 - with the liver, lungs, and skin (6/32 for both !!!!! )

Answer: thank you so much for this suggestion. We have adjusted following this recommendation.

References

Are references 19 and 20 the same?

  1. Chang SS, Reuter VE, Heston WDW, Bander NH, Grauer LS, Gaudin PB. Five different anti-prostate-specific membrane antigen (PSMA) antibodies confirm PSMA expression in tumor-associated neovasculature. Cancer Res [Internet]. 1999 Jul 1;59(13):3192– 617.
    8. Available from: http://www.ncbi.nlm.nih.gov/pubmed/10397265 618
    20. S.S. C, V.E. R, W.D.W. H, N.H. B, L.S. G, P.B. G. Five different anti-prostate-specific membrane antigen (PSMA) antibodies confirm PSMA expression in tumor-associated neovasculature. Cancer Res. 1999.

Answer: We are sorry or this confusion, the reviewer is totally right. We modified reference 20 to Tang et al., 2015.

Conclusions

The conclusions presented are important, but they are not in agreement with the objective of the work, which is to evaluate the prognostic value of certain markers.

Answer: This is a really important point, and we are so grateful for the reviewing to highlighting this point. We have adjusted and included conclusions directly related to our main goal. Thank you so much. This new information is available in lines 536-539.

Note: Supplementary files were not made available to the reviewer.

Answer: we are really sorry for this mistake. We have included the supplementary material in the revised version.

Reviewer 3 Report

The manuscript provides straightforward assessment of several potential prognostic markers for canine splenic hemangiosarcoma.  The paper is generally well written and should be of interest to readers of Animals. The authors should consider the following points:

  1. In the Introduction (Lines 51-52) it is stated, "few advances have been achieved in the past year regarding prognosis and treatment ..." The authors should include the recent paper describing immunotherapy for metastatic canine hemangiosarcoma (Lucroy et al., Evaluation of an autologous cancer vaccine for the treatment of metastatic canine hemangiosarcoma: a preliminary study. BMC Veterinary Research, 16:447, 2020.  In addition, you may want to consider changing "...achieved in the past year..." to "...achieved recently..."
  2. In the Methods, the authors should state what p value was used to determine significance of differences.
  3. Under Results, the authors should consider noting how many intact versus neutered/spayed animals were included.  This is probably not entirely essential, but will help the reader better understand the population from which samples were obtained.
  4. Figure 1 includes "Surgery and CT"  as a group; however, the acronym, "CT" is not defined.
  5. Lines 224-230 discuss chemotherapy, however some brief discussion of what chemotherapeutic regimens were used would be helpful.As well, Lines 500-505 provide some insight into this, however earlier description would be useful.  Line 500-501 describes separation of patients treated with anthracyline from those treated with cyclophosphamide, however there is no description of this in the Methods or Results.
  6. I find the text confusing compared to Table 2. Specifically, Lines 267-272 begin by stating there there was no statistically significant difference of PMSA between  the "capillary, cavernous, and solid histological types," yet Table 1 seems to indicate statistically significant differences between capillary x cavernous and cavernous x solid.  Please make sure that the text is consistent with the results presented in the table.
  7. Line 368-369 states that there was no significant difference of Ki67 expression and the three different histological subtypes; however, Table 2 indicates a significant difference between Capilary x solid.
  8. In the Discussion, Lines 480-481 you indicate an average survival time of 262.43 days.  It is always difficult to compare such data between studies when tumors of different stages have been included in the analysis.  Please add an appropriate comment to reflect this.
  9.  

Author Response

Reviewer 3

The manuscript provides straightforward assessment of several potential prognostic markers for canine splenic hemangiosarcoma.  The paper is generally well written and should be of interest to readers of Animals. The authors should consider the following points:

Answer: Dear reviewer, thank you so much for the nice overview of our manuscript and for providing such a positive criticism on our manuscript. We have followed all suggestions and provided a point-by-point answer for your comments. Once again, thank you.

1. In the Introduction (Lines 51-52) it is stated, "few advances have been achieved in the past year regarding prognosis and treatment ..." The authors should include the recent paper describing immunotherapy for metastatic canine hemangiosarcoma (Lucroy et al., Evaluation of an autologous cancer vaccine for the treatment of metastatic canine hemangiosarcoma: a preliminary study. BMC Veterinary Research, 16:447, 2020.  In addition, you may want to consider changing "...achieved in the past year..." to "...achieved recently..."

Answer: Dear reviewer, thank you so much for this suggestion. We included this reference as reference “6” and also changed the phrase, as suggested.

2. In the Methods, the authors should state what p value was used to determine significance of differences.

Answer: We apologize for not including this information. We determined p value significant when the value was lower than 0.05. The new information can be found in lines 183-184.

3. Under Results, the authors should consider noting how many intact versus neutered/spayed animals were included.  This is probably not entirely essential but will help the reader better understand the population from which samples were obtained.

Answer: Dear reviewer, in Brazil, spaying dogs is not a common procedure, and our population is based on non-castrated dogs. Thus, all animals in this research are not spayed. Since we agree with the reviewer that is an important point to highlight, we included this information in lines 94-95.

4. Figure 1 includes "Surgery and CT" as a group; however, the acronym, "CT" is not defined.

Answer: thank you for your comment. We are sorry for this mistake. We adjusted and included the acronym for CT. Once again, thank you.

5. Lines 224-230 discuss chemotherapy, however some brief discussion of what chemotherapeutic regimens were used would be helpful. As well, Lines 500-505 provide some insight into this, however earlier description would be useful.  Line 500-501 describes separation of patients treated with anthracyline from those treated with cyclophosphamide, however there is no description of this in the Methods or Results.

Answer: Dear reviewer, we included in methods section, statistical analysis subheading, information regarding segregating samples and also included a discussion paragraph, regarding treatment in lines 503-510, as requested.

6. I find the text confusing compared to Table 2. Specifically, Lines 267-272 begin by stating there there was no statistically significant difference of PMSA between  the "capillary, cavernous, and solid histological types," yet Table 1 seems to indicate statistically significant differences between capillary x cavernous and cavernous x solid.  Please make sure that the text is consistent with the results presented in the table.

Answer: Dear reviewer, we are sorry for this confusing section. We were talking about the ANOVA analysis, that compares the three groups. However, we agree with the reviewer that was confusing and we adjusted. The new information can be found in lines 273-276.

7. Line 368-369 states that there was no significant difference of Ki67 expression and the three different histological subtypes; however, Table 2 indicates a significant difference between Capilary x solid.

Answer: Dear reviewer, once again, this sentence was describing the ANOVA results, that compare the three groups (solid x cavernous x capillary, at the same time). However, we agree that makes the text confuse, and the information is still in the table. Thus, We adjusted for keeping the text clear. Thank you.

8. In the Discussion, Lines 480-481 you indicate an average survival time of 262.43 days.  It is always difficult to compare such data between studies when tumors of different stages have been included in the analysis.  Please add an appropriate comment to reflect this.

Answer: the statement was added in lines 489-491.